## [Transparent Peer Review file · Nature Communications]

Tailoring sodium and oxygen mixed-ion conduction in the A-site non-stoichiometric NaNbO_3 -based ceramics

Corresponding Author: Professor Zhiyong Liu

Version 0:

Reviewer comments:

Reviewer #1

(Remarks to the Author)

Reviewer #2

(Remarks to the Author)

This paper is interesting, but the subject of mixed Na/O conduction in the same ' NaNbO_3 ' perovskite lattice was already mentioned 5 years ago by these authors : Gouget, G. et al. Associating and tuning sodium and oxygen mixed-ion conduction in niobium-based perovskites. *Adv. Funct. Mater.* 30, 1909254 (2020). Reading the papers of Gouget et al. (ref 53 in this manuscript but also refs 21 and 22) , I consider that there are no original features in this paper submitted to Nature Comm, especially as there are other papers (refs 2& and 22) by the same team on the same class of materials. This paper should not be accepted in Nature Comm.

Reviewer #3

(Remarks to the Author)

Liu et al. reported the synergistic regulation of Na^+ and O_2^- ions conduction in the NaNb_3 -based ionic conductor. By changing the structures of these three types of polyhedrons, the conduction channels of Na^+ and O_2^- ions as well as electrons could be effectively regulated. This work is scientifically interesting and can be published after some revisions, and comments are listed below.

- 1.The section of temperature-dependent dielectric responses. The author mentioned two phase transition peaks here. However, NaNbO_3 has multiple phase transition characteristics within this temperature range. Are the phase transitions corresponding to these phase transition points accurate?
- 2.The author mentioned that the type of carriers can be regulated by the non-stoichiometry of A-site elements, thereby achieving the transformation between P-type and N-type solid-state ionic conductors. When there is a sodium deficiency, it behaves as an oxygen ionic conductor. However, sodium deficiency also creates sodium vacancies. Why does the sodium ion conductivity deteriorate? Please explain.
- 3.In Fig. 9d, both the samples with excess and deficiency show a sudden increase in conductivity at 2%. Please explain.
- 4.The lattice expansion phenomenon of the Na-deficient sample analyzed by TEM is opposite to the cell volume contraction analyzed by XRD. Will this affect the conclusion?

Reviewer #4

(Remarks to the Author)

This research reports the variation of the conduction mechanism in $\text{Na}_x\text{Ca}_{0.04}\text{Nb}_{0.96}\text{Zr}_{0.04}\text{O}_3$ - perovskite ceramics through A-site nonstoichiometry. By adjusting the Na content from deficient to excess (i.e., $x = 0.96-1.02$), the dominant

defect species changed, resulting in a transformation of the NbO₆ octahedral structure, from a flattened configuration in Na-deficient compositions to an obliquely elongated one in Na-excess compositions. Consequently, the dominant conduction mechanism shifted from O²⁻ ion conductivity (in Na-deficient samples), to mixed O²⁻ and electron (e⁻) conductivity (in stoichiometric samples), and finally to Na⁺ ion conductivity (in Na-excess samples).

The results are novel, and the insights obtained from this study are highly beneficial to the field of perovskite oxides. Rigorous experimental investigations and comprehensive analyses of the structural and electrical properties were conducted. However, there are some issues that need to be addressed:

1. In the introduction, it is not clearly explained why the authors specifically chose “Na_xCa_{0.04}Nb_{0.96}Zr_{0.04}O₃- ” as the base composition instead of pure NaNbO₃ or other potential NaNbO₃-based ceramics. Please provide the rationale for choosing this composition in the revised manuscript.
2. From Fig. 4, the ε'-T curves of all compositions (e.g., Na_{0.96}, Na_{0.99}, Na, Na_{1.01}, Na_{1.02}) at 1 kHz, 10 kHz, 100 kHz, and 1 MHz show good overlap from room temperature up to approximately 350 °C. Beyond this point, a sharp increase in both dielectric permittivity and dielectric loss is observed at 1 kHz and 10 kHz, most likely due to the thermally activated processes of defects at elevated temperature. However, for the Na_{0.98} composition, the ε'-T curve at 1 kHz exhibits dielectric peak that doesn't overlap with other frequencies at TP-R (and higher temperature), which is distinctly different from the behavior of the other compositions.

Furthermore, this composition shows the highest tan(δ) value, reaching approximately 30 at 600 °C, which is higher even than that of the Na_{0.96} composition. This observation appears to contradict the impedance spectroscopy results, which indicate that the Na_{0.96} composition has lower resistivity than that of the Na_{0.98} composition.

The authors are strongly encouraged to remeasure the dielectric data for the Na_{0.98} composition to confirm the accuracy and reproducibility of the results. Alternatively, an explanation should be provided to address this discrepancy.

3. In Figure 7, it is highly recommended to include an expanded view of the high-frequency region to allow readers to clearly observe the semicircular arc corresponding to the grain response.

4. The chemical formula “Na_{0.96x}Ca_{0.04}Nb_{0.96}Zr_{0.04}O₃- ” should be changed to “Na_xCa_{0.04}Nb_{0.96}Zr_{0.04}O₃- ” to be consistent throughout the manuscript.

Version 1:

Reviewer comments:

Reviewer #1

(Remarks to the Author)

The authors have made a good effort to appease my concerns about the stoichiometry and use of CaZrO₃ (although the formula should be Na_x not Na_{0.96x} for X =0.96 to 1.02). The structural analysis is also much clearer however the presentation and interpretation of the electrical data by DRT and IS is still not convincing for me. The first point is that Figures 5 and 6 are too small to read easily and contain too much information. Secondly there is a good fit between DRT and IS response (as shown in Fig 5 (a) based on open blue circles (IS) and red line (DRT)); HOWEVER, the equivalent circuit used for IS is based primarily on 5 RC elements but there are seven peaks in DRT so this cant be described as a good fit (if the expectation is that each RC element gives rise to a peak in DRT). R4C4 is indicated to give rise to a double peak in DRT and there is a peak at 10⁻¹s in DRT that has no corresponding RC element. Thirdly, surely to compare the quality of DRT agreement to IS response would be to plot Arrhenius plots log(tau) vs 1/T for the tau values from DRT in Fig 5 e to show the activation energies agree with those from IS in Fig 6. Finally, the insets in figure 6 are very small and its not clear to me what intercepts have been used to extract the values used in the Arrhenius plots. This is a shame as there is merit in this script but the lack of clarity in the electrical data makes it (in my opinion) not worthy of publishing until this is sorted correctly. In my previous report I suggested plotting the various capacitance values from the IS data versus temperature to try and assist with the assignment of the various arcs. The authors have not done this and I think this was an omission on their part. As it stands, this script isnt worthy for publication in Nat Comms based on the presentation and analysis of the electrical data.

Reviewer #3

(Remarks to the Author)

All issues have been satisfactorily addressed, and the manuscript is now suitable for publication.

Reviewer #4

(Remarks to the Author)

The authors have thoroughly addressed the comments raised by the reviewer. The inclusion of additional experiments, in-depth discussions, and relevant references has substantially strengthened the manuscript. I strongly recommend acceptance of this manuscript in its current form.

Version 2:

Reviewer comments:

Reviewer #1

(Remarks to the Author)

The authors have done a good job this time in responding to previous comments about IS and DRT analysis. The figures in the script are clearer and the additional SI on the analysis is useful.

RESPONSE TO REVIEWERS' COMMENTS

It is my great pleasure for your constructive comments and suggestions on our manuscript. All revised contents were marked in **red** in the revision. Our responses to the comments are as follows:

Reviewer #1 (Remarks to the Author(s)):

Although this script contains some interesting results there are many inconsistencies and a severe lack of rigour when it comes to presenting the data and providing information on how it was analysed. I outline my response in more details below and on this basis I have to suggest the script is rejected.

(i) Composition and nomenclature are confusing. No explanation as to why the starting materials include 4% Ca and Ti on the A and B-sites of the lattice, respectively. Also If we start from the given formula $\text{Na}_x\text{Ca}_{0.04}\text{Nb}_{0.96}\text{Zr}_{0.04}\text{O}_{3-d}$ then for $x=0.96$ we have a stoichiometric perovskite ABO_3 , yet this is reported as being an Na-deficient sample with Na and O vacancies and this is very confusing. The authors refer (I think) to $x=1$ being stoichiometric yet this would give a formula of $\text{A}_{1.02}\text{BO}_{3.02}$ and infer either an excess of A-site cations and oxygen ions (or conversely B-site vacancies). Taking the A and O excess model, this begs the question if ABO_3 perovskites are based on close packing of AO_3 layers how can we achieve an excess of A and O sites when no interstitial sites should be available in such a close packed lattice! (Presumably the lattice is therefore B-site deficient). I appreciate that small changes in stoichiometry associated with A/B ratios can have a dramatic effect on the electrical properties of perovskites but this requires rationalisation within the context of the crystal lattice. Late on in the script the authors suggest the presence of the A-site and O-vacancies in $x=0.96$ is associated with volatilisation of Na_2O . This should come much earlier in the script, otherwise, there is substantial confusion about the changes in composition. It might be better for the authors to frame the changes in composition on a solid solution based on a combination of $\text{Na}_{1-x}\text{NbO}_3$ and CaZrO_3 (at a level of 4 mol%) perovskites and introduce the fact that there is known non-stoichiometry in NaNbO_3 and that there is

additional issues with Na₂O loss during ceramic processing. This would clarify several issues at the outset regarding compositional design and give the reader a clear steer before the results are presented.

(ii) Structural data from laboratory XRD. There is insufficient information given pertaining to how the Rietveld refinements (RR) of the data were performed and of the results obtained. This is important because the positions of the oxygen ions are critical to the arguments about the shape of the NbO₆ octahedra and how that links to ion migration. Furthermore, where do the interstitial Na ions reside and what are the atomic positions? In the methods section it is therefore important that more information on the procedure and strategy used in the RR of the data are provided. I assume ion sites were set to be fully occupied but what were the final (x, y, z) atomic positions of all the ions and what about the isothermal parameters; were they refined or fixed and what are the values? This is important data and must be included in the results. This is especially the case given that XRD is relatively insensitive to light atoms such as O and Na, yet much is made of the final positions of these ions. Furthermore, there are no errors associated with the values given in Table S1 and these need to be included. There is a typographical error in the caption for Fig S3. It should be Na-O not Nb-O.

(iii) Microscopy. I'm not an expert in TEM but this was convincing (to me at least) that $x = 0.96$ contained more Na vacancies than $x = 1.0$ and the XPS showed there were variations in oxygen content (both Fig 3). Given the heterogeneity in the electrical behaviour (next section) of these materials and the interpretation provided, it would be good to show some SEM/EDX of the materials to establish the presence of any core-shell distributions within the samples and also EDX profiles across grains to see any variations in composition at the grain boundaries. This intermediate level of microscopic elemental analysis at the micron level is missing and needs to be included to aid the interpretation of the electrical data.

(iv). The electrical characterization was extensive and performed by Dielectric Spectroscopy (DS), impedance spectroscopy (IS) using DRT and variable P_{O2} and ion blocking measurements with YSZ; however, in many cases there was a lack of rigour

(eg DRT, more later), it was often difficult to see the ‘bulk’ response in the Z^* plots based on IS (Figs 5,6,7) and the interpretation between DRT and IS seems flawed, the P_{O_2} analysis is confusing and there is no treatment of the electronic contribution in the extraction of the transport numbers for O and Na in equations 4 and 5. I find the DS data interesting but the authors should indicate if there is any change in T_{max} associated with the P-R transition across the series of samples as this is a major event in these materials. They should also indicate what frequency was used in extracting the temperature for the T to U transition. There are considerable space charge effects at these temperatures and this may influence the values reported. The IS data are poorly presented, not well analysed and I am not convinced by the interpretation. It is very difficult to see the bulk arc (called the first arc in the script) at high frequencies in any of the Z^* plots yet this should be central to the analysis. Figures should be expanded to show it in more detail and the extracted R and C values for it presented as a function of temperature instead of a single analysis at 500 C. The extracted capacitance for this arc for 500 C is given as 4 pF/cm which corresponds to a permittivity of $\sim 40-50$. This is an order of magnitude below the 300 -500 observed in the DS data at 500 C in Fig 4. Given that the materials remain ferroelectric at 500 C, then a capacitance value of 4 pF/cm (no errors on this value provided) for the bulk is too low and physically unreasonable for $NaNbO_3$ -based materials. The authors should plot out extracted C values across the full data range and they should show some temperature dependence. By tracking this onto the DRT (Fig 5 c and 6c) with a time constant of 10^{-6} s that is associated with this grain response, it appears not to be thermally activated. It remains at this value between 400 to 600 C and this means that the ‘assigned’ bulk conductivity is not thermally activated!! This cannot be correct. The larger second arc (Fig 5) at 500 C is deconvoluted into two components that are assigned as grain boundary contributions based on C values of 0.17 and 0.14 nF/cm (no errors or showing of the fits based on the equivalent circuit, again illustrating poor rigour in the analysis). These C values correspond to permittivity of ~ 1500 to 2000 which are high but are more consistent with ferroelectric bulk responses as opposed to grain boundary responses.

The authors need to plot the temperature dependence of the extracted C values to establish if they are bulk-like (and presumably temperature dependent) or grain boundary-like (presumably with little temperature dependence). These elements are temperature dependent based on the DRT data and as the authors point out show significant changes at 540 C where there is a phase transition. Surely, it makes more sense that these elements associated with the large second arc are bulk (possibly core and shell?) rather than the grain boundaries? A full analysis of extracted R and C values across the full temperature range is required to ensure the analysis is robust and the interpretation is consistent with the physical processes occurring within the materials as they undergo phase changes. At this stage, its not worth interpretating the Po₂ data until the basic IS data are fully analysed.

(v). The DRT is simply provided in a figure and no indication of any fitting parameters are given-again illustrating poor rigour in information pertaining to data analysis.

Figure 9 based on transport numbers versus x makes some sense to me (at least for O). For $x=0.96$ there is sufficient Na₂O loss to create the level of oxygen vacancies that dominate the conductivity and the presence of this soda loss is clear from TEM/XPS data. For $x=1.0$ and 1.02 (which actually have excess Na based on the original formula), the loss of soda probably still occurs but samples have much reduced levels of vacancies and therefore are close to being stoichiometric ABO₃ with very low levels of vacancies and therefore ionic transport is low. The authors suggest that the sodium ion transport number is high in Figure 9 for $x=1.02$ but this might be linked to the lack of treatment of the electronic contribution to the conductivity used in equation 5 to estimate the transport number for Na. I also refer back to my comments linked to composition in perovskites, where are the interstitial sites for A-site cations in the ABO₃ lattice and can they located via diffraction (no evidence in this script)? I therefore find no-compelling evidence for high levels of Na ion transport in these materials. In addition, the authors often cite NASICON as an example where there is high Na ion migration. This is true but they are based on framework structures where there are vacant sites and its possible

to use this sites to create interstitial ion conduction via selective doping.

Response to Reviewer #1 comments:

Thank you very much for your comments and suggestions.

(i) It is well known that two different phase structures exist in pure sodium niobate, antiferroelectric *Pbma* phase and ferroelectric *P2₁ma* phase. This co-existence complicates the analysis of the relationship between the crystal structure and electrical conductivity. To address this challenge, we introduced a minor amount of calcium zirconate (CaZrO_3) as a dopant. This doping strategy effectively stabilizes the antiferroelectric *Pbma* phase, ensuring that NaNbO_3 adopts a single-phase structure at room temperature and reducing the influencing factors on electrical conductivity analysis (*J. Mater. Chem. C* 2020, 17, 5681-5691, *Dalton Trans.* 2015, 44, 10763-10772 and *Appl. Phys. Lett.* 2015, 107, 112904). **We also added this point in the revised manuscript.**

We sincerely appreciate the valuable suggestions put forward by the reviewers. We modified the formula from $\text{Na}_x\text{Ca}_{0.04}\text{Nb}_{0.96}\text{Zr}_{0.04}\text{O}_{3-\delta}$ to $\text{Na}_{0.96x}\text{Ca}_{0.04}\text{Nb}_{0.96}\text{Zr}_{0.04}\text{O}_{3-\delta}$ to better reflect the design of the A-site non-stoichiometry. Through detailed analysis of the crystal structure, it is believed that excess Na^+ ions are located in the Na-O-Na and Na-O-Nb spatial networks. The designed Na-excess samples exhibit a significant enhancement in electrical conductivity and remarkable Warburg behavior characteristics of ionic conduction. The test results of the sandwich-structured samples confirm the presence of Na^+ ionic conduction (as shown in Fig. 7 a and b). Subsequently, the contributions of electronic and ionic conductivities are determined by fitting the electronic conductivity and ionic conductivity using an equivalent circuit.

Fig. 7 (a) Schematic illustrations and photos of the sandwich structure, SEM image of the interface between the layers and EDS spectrum showing the elemental composition at the interface. Impedance spectra obtained from the conventional structure and sandwich structure for (b) NCNZ- $\text{Na}_{0.96}$, (c) NCNZ- $\text{Na}_{1.0}$, and (d) NCNZ- $\text{Na}_{1.02}$ samples, respectively. (e) Bar chart showing the Na^+ and O^{2-} ion conductivities of NCNZ- Na_x samples. (f) Schematic cross-sectional illustration of the Na^+ and O^{2-} ion channels, along with the Na-O-Na and Na-O-Nb networks.

Here, we attempt to emulate the more complex models using a simple equivalent circuit composed of three parallel rails (Fig. S10a). At high frequencies, the ionic rail is conductive and the total resistance is equal to the parallel combination of the electronic ($R_{\text{elec}} = d/\sigma_{\text{elec}} A$) and ionic ($R_{\text{ion}} = d/\sigma_{\text{ion}} A$) components (where σ is conductivity, d is sample thickness, and A is area). Along with the dielectric capacitance, $C_{\text{Bulk}} = \epsilon A/d$, this generates the high frequency semicircle in the Z^* plot (Fig. S10b). At low frequencies the ionic carriers have time to accumulate at the electrodes and, in the dc limit, the impedance approaches R_{elec} (*PCCP* 2001, 3, 1668-1678 and *Appl. Phys. Lett.* 2010, 96, 052906). Based on the standard model in Fig. S10b:

$$R_{\text{ion+elec}} = (\sigma_{\text{ion}} + \sigma_{\text{elec}})^{-1} \quad (1)$$

$$R_{\text{elec}} = (\sigma_{\text{elec}})^{-1} \quad (2)$$

$$t_{elec} = \frac{\sigma_{elec}}{\sigma_{total}} = \frac{R_{ion+elec}}{R_{elec}} \quad (3)$$

Here t quantifies the contribution of an individual ion to the total conductivity. R is the intercept of the fitted curve with the abscissa; σ is the electrical conductivity. According to the calculation results, the contribution of t_{elec} is approximately 8%.

Fig. S10 (a) An effective equivalent circuit used to model mixed conduction with ion, (b) simulated impedance spectra. (c) Actual curve and fitted curve based on the test curve.

These results collectively support the existence of conductive Na^+ ions in the lattice. Since the excess of Na^+ ions is relatively small (only $2\% \times 0.96$) and it is impossible to determine the position of Na^+ ions in the ABO_3 -type structure through refinement, we chose to quantify the space of the lattice interstices to prove the possible existence of interstitial Na^+ ions.

Rietveld refinement data clearly indicate that the Na-O-Na and Na-O-Nb networks exhibit expansion, as depicted in Fig. S6. Compared with the $\text{Na}_{1.0}$ sample, all the edge lengths of the Na-O-Na network in the $\text{Na}_{1.02}$ sample exhibit elongation. This indicates the expansion of the Na-O-Na network and an increase in its interstitial space. Meanwhile, the Na-O-Nb network also shows a tendency towards enlargement. The alterations in the interstitial spaces of these networks within the conductive plane are illustrated in Fig. 7f. Upon increasing the Na content from 1.0 to 1.02, notable changes occurred in the cross-sectional interstitial distances. In the Na-O-Na network, the $d(\text{Na-Na})$ distance expands from 3.894 \AA to 3.896 \AA , and the $d(\text{O-O})$ distance increases from 4.482 \AA to 4.489 \AA . Meanwhile, in the Na-O-Nb network, the $d(\text{Na-O})$ distance grows from 3.046 \AA to 3.233 \AA , while the $d(\text{Na-Nb})$ distance increases from 3.378 \AA to 3.379 \AA . This expansion of the Na-O-Na and Na-O-Nb networks serves to diminish the

migration energy barrier for Na⁺ ions.

Fig. S6 The Na–O–Na and Na–O–Nb networks of the NCNZ-Na_{1.0} and NCNZ-Na_{1.02} samples.

Table S3 Ionic radii of Na and O ions.

Ions	CN	r/Å
Na ⁺	4	0.99
	6	1.02
	8	1.18
	12	1.39
O ²⁻	2	1.21
	4	1.38
	6	1.40

Based on the above insights, the spaces of the Na-O-Na and Na-O-Nb networks are quantified to observe the possibility of the existence of interstitial ions. The

introduction of interstitial ions alters the local environment of surrounding ions, thereby changing their coordination numbers. Consequently, when quantifying the spatial characteristics of the Na-O-Na and Na-O-Nb networks, we calculated the network spaces under various coordination numbers (Table S3) to assess the possibility of the existence of interstitial ions:

Na-O-Na network:

$$i_{\text{Na-Na}} = d_{(\text{Na-Na})} - 2r_{\text{Na}} \quad (1)$$

$$i_{\text{O-O}} = d_{(\text{O-O})} - 2r_{\text{O}} \quad (2)$$

Na-O-Na network:

$$i_{\text{Na-O}} = d_{(\text{Na-O})} - r_{\text{Na}} - r_{\text{O}} \quad (3)$$

Here, i represents the interstitial space between two ions, d represents the inter-atomic spacing, and r represents the ionic radius. As shown in Fig. S6, the Na-O-Nb network is mainly restricted by the $d(\text{Na-O})$ interstices. Therefore, only the Na-O interstices are quantified here. The calculations show that the maximum value of $i(\text{Na-Na})$, which is 1.878 Å, is greater than the minimum value of $r(\text{Na})$ 0.99 Å. Similarly, the maximum value of $i(\text{O-O})$, 1.098 Å, is also greater than the minimum value of $r(\text{Na})$ 0.99 Å. Moreover, the values of $i(\text{Na-O})$ range from 1.032 Å, which is larger than the minimum value of $r(\text{Na})$ 0.99 Å. This evidence confirms the existence of interstices in the Na-O-Na and Na-O-Nb structures that allow the passage of Na ions.

In summary, it is determined that there are ion channels in the perovskite-type structure framework of NaNbO_3 for Na^+ ions to pass through, as shown in Fig. S6. The significant Na^+ ion conductivity behavior when there is an excess of Na^+ ions at the A-site is caused by interstitial ion conductivity. **We added these results and discussions to the revised manuscript.**

Regarding the issue of Na_2O volatilization, we adopted the buried sintering method to avoid volatilization when sintering the samples. Therefore, volatilization will not be a significant influencing factor in this study. To avoid confusing readers, we have deleted the content about Na_2O volatilization in the main text to ensure logical clarity.

(ii) We sincerely appreciate the valuable suggestions put forward by the reviewers. In response, we have supplemented the specific Rietveld data and presented them in Table S1 and S2. Meanwhile, we have corrected the errors in Fig. S5. The refinement was carried out using the GASA software with the *Pbma* space group, and the error values are all within the ideal range. Since the content of interstitial Na ions is relatively low (only $2\% \times 0.96$), the positions of interstitial Na ions cannot be clearly determined from the Rietveld data. We believe that the excess Na⁺ ions may be located in the Na-O-Na and Na-O-Nb networks (Fig. S6 and Table S3). The designed Na-excess samples exhibit a significant enhancement in electrical conductivity and remarkable Warburg behavior characteristics of ionic conduction. The test results of the sandwich-structured samples confirm the presence of Na⁺ ionic conduction (as shown in Fig. 7). Subsequently, the contributions of electronic and ionic conductivities are determined by fitting the electronic conductivity and ionic conductivity using an equivalent circuit (Fig. S10). After fitting calculations, ionic conductivity dominates with a value of 92%. Meanwhile, the interstitial spaces of the two networks are quantified, which shows the possibility of allowing Na⁺ ions to pass through. Therefore, interstitial Na⁺ ions exist in these two network structures (the Na-O-Na and Na-O-Nb networks). **We added these results and discussions to the revised manuscript.**

Fig. S5 Distortion of Na-O environments in NCNZ-Na_x (x = 0.96, 1 and 1.02) ceramics, respectively.

Table S1. Unit-cell parameters calculated from the Rietveld profiles of NCNZ-Na_x ceramics

Compound reference	Na _{0.96}	Na _{0.98}	Na _{0.99}	Na	Na _{1.01}	Na _{1.02}
Crystal system	Orthorhombic					
$a/\text{\AA}$	5.5605(8)	5.5610(7)	5.5618(9)	5.5619(9)	5.5626(3)	5.5669(1)
$b/\text{\AA}$	15.5572(6)	15.5574(1)	15.5614(1)	15.5617(9)	15.5621(3)	15.5678(6)
$c/\text{\AA}$	5.5006(0)	5.5018(9)	5.5024(9)	5.5025(4)	5.5025(5)	5.5069(1)
Unit cell volume/ \AA^3	475.84(4)	475.99(8)	476.23(8)	476.26(7)	476.33(8)	477.25(6)
Space group	Pbma					
Reduced χ^2	1.70	1.79	1.97	2.09	1.51	1.93
R _{wp} (%)	5.82	5.73	4.25	4.82	2.11	5.55
R _p (%)	6.45	7.95	6.73	8.24	6.01	8.47

Table S2. Crystal structure parameters for the NCNZ-Na_x ($x = 0.96$ 1.0 and 1.02)

	Site	Atom	x/a	y/b	z/c	Occ.	U _{iso}
Na _{0.96}	4c	Nb	0.2644(5)	0.1258(1)	0.2430(2)	0.96(-)	0.002(3)
	2a	Na1	0.75(-)	0(-)	0.2570(1)	0.9216(-)	0.023(6)
	2b	Na2	0.7803(4)	0.25(-)	0.2540(5)	0.9216(-)	0.038(5)
	2a	O1	0.25(-)	0(-)	0.2951(8)	0.9808(-)	0.022(8)
	2b	O2	0.2340(7)	0.25(-)	0.1711(5)	0.9808(-)	0.011(4)
	4c	O3	0.0270(3)	0.1428(6)	0.5241(9)	0.9808(-)	0.010(1)
	4c	O4	0.4580(1)	0.1108(7)	-0.0449(4)	0.9808(-)	0.015(3)
	4c	Zr	0.2644(5)	0.1258(1)	0.2430(2)	0.04(-)	0.029(6)
	2a	Ca1	0.75(-)	0(-)	0.2570(1)	0.04(-)	0.080(1)
	2b	Ca2	0.7803(4)	0.25(-)	0.2540(5)	0.04(-)	0.068(4)
Na _{1.0}	4c	Nb	0.2624(8)	0.1256(1)	0.2436(5)	0.96(-)	0.004(2)
	2a	Na1	0.75(-)	0(-)	0.2558(5)	0.96(-)	0.010(4)
	2b	Na2	0.7785(4)	0.25(-)	0.2528(5)	0.96(-)	0.010(4)
	2a	O1	0.25(-)	0(-)	0.3017(5)	1(-)	0.007(4)
	2b	O2	0.2419(7)	0.25(-)	0.1777(5)	1(-)	0.007(4)
	4c	O3	0.0349(7)	0.1360(1)	0.5307(5)	1(-)	0.007(4)
	4c	O4	0.4658(1)	0.1038(1)	-0.0385(5)	1(-)	0.007(4)
	4c	Zr	0.2624(8)	0.1256(1)	0.2436(5)	0.04(-)	0.008(9)
	2a	Ca1	0.75(-)	0(-)	0.2558(5)	0.04(-)	0.008(3)
	2b	Ca2	0.7785(4)	0.25(-)	0.2528(5)	0.04(-)	0.008(3)
Na _{1.02}	4c	Nb	0.2645(6)	0.1254(1)	0.2447(5)	0.96(-)	0.004(7)
	2a	Na1	0.75(-)	0(-)	0.2562(5)	0.96(-)	0.006(4)
	2b	Na2	0.7809(4)	0.5(-)	0.2532(5)	0.96(-)	0.023(8)
	2a	O1	0.25(-)	0(-)	0.3035(5)	1(-)	0.006(2)
	2b	O2	0.2395(5)	0.5(-)	0.1823(5)	1(-)	0.004(6)
	4c	O3	0.0325(5)	0.1434(1)	0.5353(5)	1(-)	0.019(7)
	4c	O4	0.4627(1)	0.1088(9)	-0.0331(6)	1(-)	0.006(7)

4c	Zr	0.2645(6)	0.1254(1)	0.2447(5)	0.04(-)	0.027(4)
2a	Ca1	0.75(-)	0(-)	0.2562(5)	0.04(-)	0.026(4)
2b	Ca2	0.7809(4)	0.5(-)	0.2532(5)	0.04(-)	0.026(4)

(iii) We sincerely appreciate the valuable suggestions put forward by the reviewers. In response, we conducted a more detailed observation of the morphology and elemental distribution of the samples. The scanning electron microscopy (SEM) images of the non-stoichiometric and stoichiometric samples, as shown in Fig. S1, reveal a uniform distribution of grains. Notably, no obvious secondary phases are observed within the grains. To conduct a detailed examination of the internal structure and elemental distribution of the grains, transmission electron microscopy (TEM) analysis was carried out. We prepared ion thinning samples and analyzed the morphology and elemental distribution at the grain positions using TEM and EDS, as shown in Fig. S2. The analysis results indicate that the elements are uniformly distributed inside the grains, and there are no characteristics of a second phase or core-shell structure. Besides, a comprehensive review is conducted on the structural research of NaNbO₃-based ceramics with similar compositions. No reports are found regarding the presence of core-shell structures in the components of such materials (*J. Euro. Ceram. Soc.* 2018, 38, 4939–4945, *J. Appl. Phys.* 2015, 118, 174107, and *Acta Mater.* 2018, 161, 352-359).

Fig. S2 transmission electron microscopy (TEM) analysis reveals the morphology images and corresponding energy dispersive X-ray spectroscopy (EDS) elemental distribution maps.

(iv) We sincerely appreciate the valuable suggestions put forward by the reviewers. In response, we additionally measured the XRD patterns (Fig. 4) and dielectric spectra from room temperature to 700 °C to jointly characterize the phase transition of the NCNZ- Na_x samples. The analysis of the phase transition in the dielectric spectra mainly relies on the data at 10 kHz. By comparing with the temperature-dependent XRD patterns, we aim to determine the influence law of the phase transition on the electrical conductivity performances.

Fig. 4 (a) XRD patterns and structural transitions of NCNZ- $\text{Na}_{0.96}$ with various temperatures. (b-g) Dielectric constant and loss of the NCNZ- Na_x ceramics as a function of temperature.

After replacing the electrode materials, we re-measured the impedance spectra (Fig. 5) of each sample and plotted the capacitance (C') curves as functions of temperature (T) and frequency (f) (Fig. S7b). Herein, the fitting method of DRT is as follows:

Distribution of Relaxation Times

The method of the distribution of relaxation times (DRT) allows a separation of electrochemical processes with different time constants which can be calculated from

measured impedance data. Thus, it allows a clearer look on the measured spectra and helps to resolve processes which might not be detected otherwise. The numerical scheme follows a Levenberg-Marquardt algorithm. This is based on the fact that impedance functions obeying the Kramers-Kronig relations (KK) can be represented as a number of infinitesimal differential RC-elements. The underlying theory is discussed in detail in various references such as *Electrochim. Acta* 2017, 228, 652-658.

$$Z(\omega) = R_0 - R_{pol} \int_0^{\infty} \frac{\gamma(\tau)}{1 + j\omega\tau} d\tau \quad (7)$$

Given the relaxation time distribution $g(t)$ is defined as:

$$\int \gamma(\tau) d\tau = \sum_{i=1}^{\infty} R_i \quad (8)$$

This classical approach has the drawback that data preprocessing is required to remove the high-frequency region influenced by the cable's inductance and the low-frequency region representing the diffusion limitation. To better meet the requirements of the impedance spectra, the general approach of Eq. (7) is expanded with an inductance L , mirroring the imaginary part of the impedance at high frequencies. A capacity contributing to the impedance spectrum at low frequency, the ohmic resistance R_0 causes by the bulk resistance of the electrolyte, a Warburg element and a capacitor representing the diffusion processes. The Warburg element represents the diffusion behavior for particles in micron range whereas the capacitor is added to model the diffusion smaller particles in the nanometer range. Thus, the measured impedance data are fitted to the following equation:

$$Z_{modle}(\omega) = j\omega L - \frac{1}{j\omega C} - \frac{A_W}{\sqrt{\omega}}(1 - j) + R_0 - R_{pol} \int_0^{\infty} \frac{\gamma(\tau)}{1 + j\omega\tau} d\tau \quad (9)$$

In its discretized form the distribution integral can be expressed by a matrix multiplication and thus becoming a system of linear equations and adopting the form $y=Ax$:

$$Z_{modle}\tau = \begin{bmatrix} 1 & \dots & 1 \\ \frac{1}{1 + (\omega\tau_1)^2} & \dots & \frac{1}{1 + (\omega\tau_n)^2} \\ \vdots & \ddots & \vdots \\ 1 & \dots & 1 \\ \frac{1}{1 + (\omega m\tau_1)^2} & \dots & \frac{1}{1 + (\omega m\tau_n)^2} \end{bmatrix} \cdot \gamma + j \begin{bmatrix} \frac{\omega\tau_1}{1 + (\omega\tau_1)^2} & \dots & \frac{\omega\tau_n}{1 + (\omega\tau_n)^2} \\ \vdots & \ddots & \vdots \\ \frac{\omega m\tau_1}{1 + (\omega m\tau_1)^2} & \dots & \frac{\omega m\tau_n}{1 + (\omega m\tau_n)^2} \end{bmatrix} \quad (10)$$

Details of the used numerical routine are described in.

Due to the duration time and drifts in the system unwanted artifacts especially at lower frequencies might occur. To ensure that the measured impedance data match the KK-relation, they are transformed with the Z-HIT function which calculates the impedance data $Z(f)$ from the phase data $\varphi(\omega)$ after the following approximation:

$$\ln|H(\omega_0)| \approx const. + \frac{2}{\pi} \int_{\omega_s}^{\omega_0} \varphi(\omega) d\ln\omega + \gamma \frac{d\varphi(\omega)}{d\ln\omega} \quad (11)$$

The fitting results of DRT are shown in Fig. 5a. To further ensure the accuracy of DRT analysis, the $C'-f$ plots at several temperatures are displayed in Fig. S7b. The high-capacitance plateau corresponds to the grain boundary response, the low capacitance plateau corresponds to the grain response, and both grain and grain boundaries exhibit temperature dependence. The measured capacitance responses are in good agreement with the fitting values, validating the rationality of the distribution of relaxation times (DRT) analysis.

Fig. 5 (a) Matching parts of the impedance spectra with the corresponding components of the DRT

function. (b) Nyquist plots of the NCNZ- Na_x ceramics at 500 °C and (c) the corresponding DRT diagram. (d) The Nyquist plots of NCNZ- $\text{Na}_{0.96}$ samples at different temperatures, (e) normalized DRT curves.

Fig. S7 (a) DRT curves of the NCNZ- $\text{Na}_{0.96}$ sample measured at various temperatures. (b) The C' spectra for the NCNZ- $\text{Na}_{0.96}$ sample.

Fig. S8 (a and b) The Nyquist plots of NCNZ- $\text{Na}_{1.0}$ and NCNZ- $\text{Na}_{1.02}$ samples, (c and d) normalized DRT curves at different temperatures.

As shown in Fig. 5 and Fig. S8, the semi-circle representing the grain boundary response shifts towards higher frequencies as the temperature increases, whereas the semi-circle corresponding to the grain response gradually vanishes with rising temperature. This behavior is consistent with the response peaks of grains and grain boundaries in the DRT analysis, which can be attributed to thermal activation. However, due to the high degree of overlap between the grain boundary and grain response semi-circles and the pronounced hindering effect from the grain boundaries, the semi-circle of the grain response at high frequencies is rather weak. Therefore, the DRT analysis offers a more intuitive way for observation. Additionally, Fig. S1 and S2 confirm the absence of a core-shell effect. **We added these results and discussions to the revised manuscript.**

(v) To quantitatively analyze the contributions of ionic and electronic conduction and determine the types of charge carriers related to conduction, we fitted the electronic and ionic conduction contributions of $\text{Na}_{1.02}$ using the circuit and model shown in Fig. S10a and b. The fitting results show that the contribution of electronic conductivity is only 8%, as presented in “Analysis of the contributions of ionic and electronic conduction” below. Therefore, we neglected the contribution of electronic conduction in the main manuscript.

Analysis of the contributions of ionic and electronic conduction

Fig. S10 (a) An effective equivalent circuit used to model mixed conduction with ion, (b) simulated impedance spectra. (c) Actual curve and fitted curve based on the test curve.

Here, we attempt to emulate the more complex models using a simple equivalent circuit composed of three parallel rails (Fig. S10a). At high frequencies, the ionic rail is conductive and the total resistance is equal to the parallel combination of the electronic ($R_{elec} = d/\sigma_{elec} A$) and ionic ($R_{ion} = d/\sigma_{ion} A$) components (where σ is conductivity, d is sample thickness, and A is area). Along with the dielectric capacitance, $C_{Bulk} = \epsilon A/d$, this generates the high frequency semicircle in the Z^* plot (Fig. S10b). At low frequencies the ionic carriers have time to accumulate at the electrodes and, in the dc limit, the impedance approaches R_{elec} (PCCP 2001, 3, 1668-1678 and *Appl. Phys. Lett.* 2010, 96, 052906). Based on the standard model in Fig. S10b:

$$R_{ion+elec} = (\sigma_{ion} + \sigma_{elec})^{-1} \quad (1)$$

$$R_{elec} = (\sigma_{elec})^{-1} \quad (2)$$

$$t_{elec} = \frac{\sigma_{elec}}{\sigma_{total}} = \frac{R_{ion+elec}}{R_{elec}} \quad (3)$$

Here t quantifies the contribution of an individual ion to the total conductivity. R is the intercept of the fitted curve with the abscissa. σ is the electrical conductivity. According to the calculation results, the contribution of t_{elec} is approximately 8%.

The significant contribution of Na^+ ions to conduction is demonstrated through calculations (Fig. S10c). The expansion of the two Na-O-Na and Na-O-Nb networks (Fig. S6 and Table S3), the presence of Na-ion conduction, and the weakening of oxygen ion conduction are sufficient to prove that the conduction type when there is an excess of Na is interstitial Na-ion conduction.

Reviewer: #2 (Remarks to the Author(s):

This paper is interesting, but the subject of mixed Na/O conduction in the same ‘ NaNbO_3 ’ perovskite lattice was already mentioned 5 years ago by these authors: Gouget, G. et al. Associating and tuning sodium and oxygen mixed-ion conduction in niobium-based perovskites. *Adv. Funct. Mater.* 30, 1909254 (2020). Reading the papers of Gouget et al. (ref 53 in this manuscript but also refs 21 and 22), I consider that there are no original features in this paper submitted to Nature Comm, especially as there are

other papers (refs 2& and 22) by the same team on the same class of materials. This paper should not be accepted in Nature Comm.

Response to Reviewer #2 comments:

Thank you very much for your comments and suggestions.

Gouget et al.'s work mainly focuses on adjusting NaNbO₃-based solid-state ion conductors through doping and interlayer structure, and draws conclusions by analyzing changes in unit cell parameters. Indeed, the behavior of mixed conductivity of Na⁺ and O²⁻ ions in NaNb(Ti)O₃ ceramics was revealed. However, it is known that pure NaNbO₃ has a dual-phase (*P2₁ma* and *Pbma*) structure at room temperature, which exhibits antiferroelectric and ferroelectric characteristics respectively. This is unfavorable for analyzing the corresponding relationship between unit cell parameters and conduction forms. In this paper, we doped CaZrO₃ to stabilize the antiferroelectric phase structure (*Pbma*) of NaNbO₃, which allows for a more accurate determination of the changes in unit cell parameters corresponding to the variations in ionic conductivity. In addition, in terms of regulation methods, the non-stoichiometry at the A-site is different from the doping at the B-site. The changes in the NbO₆ octahedron induced by the occupancy rate of A-site atoms in the unit cell show more relevance to the intrinsic crystal structure of NaNbO₃ (*Adv. Funct. Mater.* 2020, 30, 2001840, and *Chem. Commun.*, 2019, 55, 2609-2612).

The exploration of the conductance behavior of perovskite solid-state ionic conductors is carried out through Nyquist curve analysis. The electrochemical response process from high frequency to low frequency includes responses from the bulk phase, grain boundaries, electrodes, etc. Currently, the main analysis method is to perform equivalent circuit fitting using Zview software. Nevertheless, some of the electrochemical response Nyquist semicircles highly overlap, making it impossible to determine the number of CP elements in the equivalent circuit, which will affect the

accuracy of the analysis. In this study, the Nyquist curve is analyzed using the distribution of relaxation times (DRT). The equivalent circuit model related to the electrochemical process is accurately measured, and the test results are specifically analyzed. By observing the area of the DRT curve and the frequency changes corresponding to the peaks, the changes in grain and grain boundary conductance under variations in composition and temperature can be effectively determined. For example, in the analysis of electrical conductivity changes, the grain response and grain boundary response in the Nyquist plot of NaNbO_3 will partially overlap, and as the temperature rises, the grain response will shift forward and gradually disappear. This is unfavorable for analyzing the conduction change form through a single Nyquist plot. DRT provides an intuitive method for analyzing the law of unit-cell parameter changes caused by the variation of electrical conductivity with composition, which is hidden under the Nyquist curve.

In distinguishing the conductivities of oxygen ions, sodium ions, and electrons, we conducted a detailed analysis of the test results from aspects such as atmosphere design, structure design, and equivalent circuit fitting. Among them, the contribution of ion conductivity and electron conductivity is distinguished by equivalent circuit fitting to ensure the reliability of experimental results. Meanwhile, a hypothesis is proposed regarding the ion channels when there is an excess of Na^+ ions and we quantify the possibility of these channels through calculations.

Based on the above analysis, this work is highly innovation and practical significance in terms of composition design, crystal structure regulation, and ionic conductivity analysis.

Reviewer: #3 (Remarks to the Author(s):

Liu et al. reported the synergistic regulation of Na^+ and O^{2-} ions conduction in the NaNb_3 -based ionic conductor. By changing the structures of these three types of polyhedrons, the conduction channels of Na^+ and O^{2-} ions as well as electrons could be

effectively regulated. This work is scientifically interesting and can be published after some revisions, and comments are listed below.

1. The section of temperature-dependent dielectric responses. The author mentioned two phase transition peaks here. However, NaNbO_3 has multiple phase transition characteristics within this temperature range. Are the phase transitions corresponding to these phase transition points accurate?

2. The author mentioned that the type of carriers can be regulated by the non-stoichiometry of A-site elements, thereby achieving the transformation between P-type and N-type solid-state ionic conductors. When there is a sodium deficiency, it behaves as an oxygen ionic conductor. However, sodium deficiency also creates sodium vacancies. Why does the sodium ion conductivity deteriorate? Please explain.

3. In Fig. 9d, both the samples with excess and deficiency show a sudden increase in conductivity at 2%. Please explain.

4. The lattice expansion phenomenon of the Na-deficient sample analyzed by TEM is opposite to the cell volume contraction analyzed by XRD. Will this affect the conclusion?

Response to Reviewer #3 comments:

Thank you very much for your comments and suggestions.

1. Sodium niobate exhibits multiple phase transitions spanning from room temperature to 700 °C, encompassing crystal structures such as *Pbma*, *Pnmm*, *Cmmm*, *P4/mbm*, and *Pm-3m*. To accurately elucidate the correlation between the phase transition behavior of the NCNZ- Na_x samples, supplementary X-ray diffraction (XRD) measurements were conducted on the NCNZ- Na_x samples over the temperature range from room temperature to 700 °C. The results are presented in Fig. 4a. Temperature dependent XRD (Fig. 4a) directly reveals phase transition behavior in NCNZ- Na_x samples. The NCNZ- $\text{Na}_{0.96}$ sample transitions from orthorhombic (*Pbma*, *Pnmm*,

Ccmm) to tetragonal (*P4/mbm*) and then cubic (*Pm-3m*) phases from room temperature to 700 °C. Unit-cell symmetry and volume changes, related to NbO₆ octahedra distortion and torsion, significantly affect the ionic conductivity. Temperature-dependent dielectric constant and loss variations disclose conductivity relaxation associated with these phase transitions. In the temperature range from 300 °C to 600 °C, the NCNZ-Na_x samples display two distinct phase transition peaks, corresponding to the transitions from the orthorhombic antiferroelectric *Pbma* (*P*) phase to the orthorhombic ferroelectric *Pnmm* (*R*) phase, and the tetragonal *P4/mbm* (*T*₂) phase to the cubic *pm-3m* (*U*) phase, respectively. Distinct phase transition peaks are indicative of substantial changes in the crystal structure. Prominently, intense phase transition peaks of *T*₂-*U* (~ 560 °C) are observed in the NCNZ-Na_x samples with increasing Na-deficiency (Fig. 4b-d). The phase transition temperature is basically consistent with the turning temperature of the conductivity change observed in Fig. 6, and it is considered to be the main factor influencing the conductivity change.

Fig. 4 (a) XRD patterns and structural transitions of NCNZ-Na_{0.96} with various temperatures. (b-g) Dielectric constant and loss of the NCNZ-Na_x ceramics as a function of temperature.

2. The refined data are imported into Diamond software to observe the local environment of the atoms. Due to the low number of O vacancies ($V_{\text{O}}^{\bullet\bullet}$) and Na vacancies (V_{Na}') in the compound, it is impossible to accurately identify oxygen and Na sites that are partially occupied. The local environments of Nb and Na ions for NCNZ- $\text{Na}_{0.96}$, NCNZ- $\text{Na}_{1.0}$ and NCNZ- $\text{Na}_{1.02}$ samples are shown in Fig. 2c and Fig. S5. For the Na-deficient samples, the average bond distances (~ 1.9940 Å) for $d(\text{Nb-O3})$ and $d(\text{Nb-O4})$ in the equatorial plane are shorter than those in the stoichiometric samples (~ 1.9941 Å), and there is a large difference between the bond distances. This is consistent with the observed reduction in the a and c lattice parameters (Table S1). Meanwhile, $d(\text{Nb-O1})$ and $d(\text{Nb-O2})$ bond distances also decrease relatively, and the change in $d(\text{Nb-O2})$ is more significant. The phenomenon suggests the emergence of flattened NbO_6 octahedra, a structural feature that is expected to facilitate the conduction of oxygen ions. However, as shown in Fig. S5, regarding the $d(\text{Na-O})$ spacing in the NaO_{12} icosahedron related to the Na^+ ion conductivity, the $\text{Na}_{0.96}$ sample exhibits an expansion trend compared with the $\text{Na}_{1.0}$ sample, showing a change opposite to that of the NbO_6 octahedron. This might be the structural reason for the decrease in the Na^+ ion conductivity. In addition, inspection of Fig. 6a and d reveals that the electrical conductivity under a nitrogen atmosphere is markedly higher than that under oxygen and air atmospheres, suggesting a strong correlation between the conductivity variation and the oxygen partial pressure. An elevation in oxygen partial pressure is found to facilitate oxygen-ion conductivity, a phenomenon intricately linked to the concentration of charge carriers. Consequently, as the gaseous environment transitions from nitrogen to oxygen, the decline in conductivity can be ascribed to the alteration in charge carrier concentration. A mutual suppression effect exists between Na^+ and O^{2-} charge carriers. **We added these results and discussions to the revised manuscript.**

3. In the crystal structures of solid-state ionic conductors, point defects play a multifaceted role. They can induce structural distortions within the crystal lattice. Simultaneously, vacancies contribute to the establishment of migration pathways for

charge carriers, thereby enhancing ionic conductivity. Nonetheless, a limited quantity of point defects can aggregate to create carrier trapping sites and defect dipoles. These structural features act as impediments to ionic conduction and are thus detrimental to the overall electrical conductivity of the material. In scenarios where the defect concentration is relatively low, analogous structures may form, impeding the flow of charge carriers and giving rise to an unexpected surge in electrical resistance or a sudden increase in a related conductivity associated parameter. However, as the concentration of defects increases, this hindering effect dissipates, and ionic channels are established. Consequently, a sharp rise in electrical conductivity is observed.

4. Transmission electron microscopy (TEM) is employed to examine the local unit cell features, whereas XRD is more suited for analyzing the overall lattice characteristics. At defect sites, the equilibrium of inter-atomic interactions surrounding vacancy positions is disrupted. Neighboring atoms, experiencing reduced inward directed binding forces, relax towards the vacancies, leading to lattice expansion at these sites. Nevertheless, local structural features can induce atomic elastic distortions. These distortions generate long range stress fields that cause an increase in the inter-atomic distances in remote regions.

In addition, a comprehensive understanding of the lattice parameters can be achieved through the refinement of XRD data. In contrast, TEM data only contain diffraction information from specific crystal planes. Both types of data are related to atomic spacing. The local environment diagrams of Na and Nb atoms obtained after XRD refinement are presented in Fig. 2 and Fig. S5. It can be observed that as the Na occupancy at the A-site decreases, the unit-cell parameters decrease. However, the asymmetry of the NbO₆ octahedra and NaO₁₂ icosahedra increases. This enhanced asymmetry leads to an increase in the local atomic spacing, consequently inducing an enlargement of the spacing of certain crystal planes. For instance, as the Na occupancy decreases, the d(Na1-O3) distance expands from 3.0465 Å to 3.1596 Å. This change is consistent with the variation in crystal plane spacing observed in the TEM images. Thus,

despite these complex local and long-range effects, they do not undermine the validity of our analysis and subsequent judgment.

Fig. S5 Distortion of Na-O environments in NCNZ- Na_x ($x = 0.96, 1$ and 1.02) ceramics, respectively.

Reviewer: #4 (Remarks to the Author(s):

This research reports the variation of the conduction mechanism in $\text{Na}_x\text{Ca}_{0.04}\text{Nb}_{0.96}\text{Zr}_{0.04}\text{O}_{3-\square}$ perovskite ceramics through A-site non-stoichiometry. By adjusting the Na content from deficient to excess (i.e., $x = 0.96-1.02$), the dominant defect species changed, resulting in a transformation of the NbO_6 octahedral structure, from a flattened configuration in Na-deficient compositions to an obliquely elongated one in Na-excess compositions. Consequently, the dominant conduction mechanism shifted from O^{2-} ion conductivity (in Na-deficient samples), to mixed O^{2-} and electron (e^-) conductivity (in stoichiometric samples), and finally to Na^+ ion conductivity (in Na-excess samples).

The results are novel, and the insights obtained from this study are highly beneficial to the field of perovskite oxides. Rigorous experimental investigations and comprehensive analyses of the structural and electrical properties were conducted. However, there are some issues that need to be addressed:

1. In the introduction, it is not clearly explained why the authors specifically chose “ $\text{Na}_x\text{Ca}_{0.04}\text{Nb}_{0.96}\text{Zr}_{0.04}\text{O}_{3-\square}$ ” as the base composition instead of pure NaNbO_3 or other

potential NaNbO₃-based ceramics. Please provide the rationale for choosing this composition in the revised manuscript.

2. From Fig. 4, the ϵ_r -T curves of all compositions (e.g., Na_{0.96}, Na_{0.99}, Na, Na_{1.01}, Na_{1.02}) at 1 kHz, 10 kHz, 100 kHz, and 1 MHz show good overlap from room temperature up to approximately 350 °C. Beyond this point, a sharp increase in both dielectric permittivity and dielectric loss is observed at 1 kHz and 10 kHz, most likely due to the thermally activated processes of defects at elevated temperature. However, for the Na_{0.98} composition, the ϵ_r -T curve at 1 kHz exhibits dielectric peak that doesn't overlap with other frequencies at TP-R (and higher temperature), which is distinctly different from the behavior of the other compositions.

Furthermore, this composition shows the highest $\tan(\delta)$ value, reaching approximately 30 at 600 °C, which is higher even than that of the Na_{0.96} composition. This observation appears to contradict the impedance spectroscopy results, which indicate that the Na_{0.96} composition has lower resistivity than that of the Na_{0.98} composition.

The authors are strongly encouraged to remeasure the dielectric data for the Na_{0.98} composition to confirm the accuracy and reproducibility of the results. Alternatively, an explanation should be provided to address this discrepancy.

3. In Figure 7, it is highly recommended to include an expanded view of the high-frequency region to allow readers to clearly observe the semicircular arc corresponding to the grain response.

4. The chemical formula “Na_{0.96x}Ca_{0.04}Nb_{0.96}Zr_{0.04}O_{3-□}” should be changed to “Na_xCa_{0.04}Nb_{0.96}Zr_{0.04}O_{3-□}” to be consistent throughout the manuscript.

Response to Reviewer #4 comments:

Thank you very much for your comments and suggestions.

1. It is well known that two different phase structures exist in pure sodium niobate,

antiferroelectric *Pbma* phase and ferroelectric *P2₁ma* phase. This co-existence complicates the analysis of the relationship between the crystal structure and electrical conductivity. To address this challenge, we introduced a minor amount of calcium zirconate (CaZrO₃) as a dopant. This doping strategy effectively stabilizes the antiferroelectric *Pbma* phase, ensuring that NaNbO₃ adopts a single-phase structure at room temperature and reducing the influencing factors on electrical conductivity analysis (*J. Mater. Chem. C* 2020, 17, 5681-5691, *Dalton Trans.* 2015, 44, 10763-10772 and *Appl. Phys. Lett.* 2015, 107, 112904). **We also added this point in the revised manuscript.**

2. We extend our heartfelt gratitude to the reviewers for their invaluable suggestions. In response, we revisited the measurements for the compositions corresponding to Na_{0.96} and Na_{0.98}. The experimental outcomes indicate that both the dielectric response and the loss curves of Na_{0.96} display notably higher values. Furthermore, the degree of overlap in the frequencies of the dielectric peaks for Na_{0.98} is consistent with that observed in the other samples (as shown in Fig. 4). To clarify the correspondence between phase transitions and dielectric response peaks, we supplemented the study with temperature dependent XRD patterns (Fig. 4a). This data directly reveals the phase transition behavior in NCNZ-Na_x samples. Specifically, upon heating from room temperature to 700 °C, the NCNZ-Na_{0.96} sample undergoes a sequential phase transition from orthorhombic (*Pbma*, *Pnmm*, *Ccmm*) to tetragonal (*P4/mbm*) and finally to cubic (*Pm-3m*) phases. The changes in unit-cell symmetry and volume, which are associated with the distortion and torsion of NbO₆ octahedra, have a substantial impact on electrical conductivity. By integrating these temperature dependent XRD data with the corresponding data on the temperature dependent dielectric constant and loss, we can precisely analyze the influence patterns of phase transition behavior as a function of composition and temperature. **We added these results and discussions to the revised manuscript.**

Fig. 4 (a) XRD patterns and structural transitions of NCNZ-Na_{0.96} with various temperatures. (b-g) Dielectric constant and loss of the NCNZ-Na_x ceramics as a function of temperature.

3. We extend our sincere gratitude to the reviewers for their invaluable suggestions. In response, we incorporated an enlarged illustration depicting the grain response within the high frequency regime into Fig. 6.

Fig. 6 The Nyquist plots of (a) NCNZ-Na_{0.96}, (b) NCNZ-Na_{1.0}, and (c) NCNZ-Na_{1.02} at various atmosphere, and (d-f) the corresponding Arrhenius diagram.

4. We extend our heartfelt gratitude to the reviewers for their insightful and constructive feedback. In light of their suggestions, we have rectified the inaccuracy in the chemical formula. Specifically, the original formula has been substituted with $\text{Na}_{0.96x}\text{Ca}_{0.04}\text{Nb}_{0.96}\text{Zr}_{0.04}\text{O}_3$.

Besides, some modifications were made in the revised manuscript.

- 1) Adequate **References** were cited in the revised manuscript.
- 2) **We polished the whole manuscript carefully again** and tried to avoid any grammatical and formatting errors. All revised contents were marked in red in the revised manuscript.

We greatly appreciate for Editors' enthusiasms and Reviewer's valuable comments, and hope that these corrections can meet them. Thank you very much.

Yours sincerely,

Zhiyong Liu

According to the suggestions by reviewer, we submit our revised manuscript with the title “**Tailoring sodium and oxygen mixed-ion conduction in the A-site non-stoichiometric NaNbO₃-based ceramics**” to your journal as attached. All revised contents were marked in **red** in the revision. Our responses to the comments are as follows:

Reviewer #1 Remarks to the Author (s):

The authors have made a good effort to appease my concerns about the stoichiometry and use of CaZrO₃ (although the formula should be Na_x not Na_{0.96x} for $x=0.96$ to 1.02). The structural analysis is also much clearer however the presentation and interpretation of the electrical data by DRT and IS is still not convincing for me.

1. The first point is that Figures 5 and 6 are too small to read easily and contain too much information.
2. Secondly there is a good fit between DRT and IS response (as shown in Fig 5 (a) based on open blue circles (IS) and red line (DRT)); However, the equivalent circuit used for IS is based primarily on 5 RC elements but there are seven peaks in DRT so this can't be described as a good fit (if the expectation is that each RC element gives rise to a peak in DRT). R4C4 is indicated to give rise to a double peak in DRT and there is a peak at 10⁻¹s in DRT that has no corresponding RC element.
3. Thirdly, surely to compare the quality of DRT agreement to IS response would be to plot Arrhenius plots $\log(\tau)$ vs $1/T$ for the τ values from DRT in Fig 5 e to show the activation energies agree with those from IS in Fig 6.
4. Finally, the insets in figure 6 are very small and it's not clear to me what intercepts have been used to extract the values used in the Arrhenius plots. This is a shame as there is merit in this script but the lack of clarity in the electrical data makes it (in my opinion) not worthy of publishing until this is sorted correctly.
5. In my previous report I suggested plotting the various capacitance values from the IS data versus temperature to try and assist with the assignment of the various arcs. The authors have not done this and I think this was an omission on their part. As it

stands, this script isnt worthy for publication in Nat Comms based on the presentation and analysis of the the electrical data.

Response to Reviewer #1 comments:

Thank you very much for your comments and suggestions.

1. We sincerely appreciate the reviewers for their insightful comments. In response, we have revised Fig. 5 and 6 to present the Nyquist curves and DRT data more clearly.

Fig. 5 **a** Matching part of the impedance spectra with the corresponding components of the DRT function. **b** Nyquist plots of the NCNZ-Na_x ceramics at 500 °C and **c** the corresponding DRT diagram. **d** The Nyquist plots of NCNZ-Na_{0.96} sample at different temperatures. **e** Partial DRT curves of the NCNZ-Na_{0.96} sample measured at various temperatures and **f** normalized DRT curves.

Fig. 6 The Nyquist plots of **a** NCNZ- $\text{Na}_{0.96}$, **b** NCNZ- $\text{Na}_{1.0}$, and **c** NCNZ- $\text{Na}_{1.02}$ samples at various atmosphere, and **d-f** the corresponding Arrhenius diagram.

2. We extend our sincere gratitude to the reviewers for their constructive comments. Guided by the frequencies of the relaxation peaks manifested in the Distribution of Relaxation Times (DRT), we designed a new equivalent circuit and conducted fitting. As shown in Fig. 5a, the first semicircle of the complex impedance spectra corresponds to the grain effect of NCNZ- Na_x ceramics. The characteristic capacitance value is $2.2458 \times 10^{-11} \text{ F} \cdot \text{cm}^{-1}$ and the frequency at the maximum value of the semicircle is $1.06 \times 10^6 \text{ Hz}$, which proves this statement. The second semicircle is composed of two arcs, each corresponding to an RC element. The characteristic capacitance of the first arc is $1.6990 \times 10^{-10} \text{ F} \cdot \text{cm}^{-1}$ and the characteristic capacitance of the second arc is $1.3857 \times 10^{-10} \text{ F} \cdot \text{cm}^{-1}$. Both of these arcs are attributed to the grain boundary effect, which is consistent with the result of the C' spectra (Fig. S7b). Temperature-dependent capacitances of grain and grain boundary (Fig. S8) align well with the dielectric spectra (Fig. 4b-g). Notably, the capacitance of grain boundary is approximately $5 \times 10^{-10} \text{ F} \cdot \text{cm}^{-1}$ and exhibit temperature independent, whereas the capacitance of grain decreases significantly as temperature increases (Fig. S8). The last polarization tail consists of four parts that can be classified into two categories: one with characteristic capacitances ranging from 3.9537×10^{-7} to $9.0655 \times 10^{-7} \text{ F} \cdot \text{cm}^{-1}$ and

frequencies from 2860 to 274 Hz. The other with characteristic capacitances spanning from 1.4754×10^{-6} to 1.3352×10^{-6} F·cm⁻¹ and frequencies from 42.8 to 8.7 Hz. The former category is associated with the ohmic contact resistance at the electrolyte/electrode interface or the high relaxation steps of the electrode fabrication process. In contrast, the latter category is ascribed to the electrode level, given its relatively larger capacitance and lower frequency. **We added these results and discussions to the revised manuscript.**

3. We are extremely grateful to the reviewers for their insightful comments. By extracting the relaxation time (τ) associated with the grain boundary peak and applying the Arrhenius equation for fitting, the activation energy E_{af} can be determined:

$$\tau = \tau_0 \exp(-E_{af}/kT) \quad (1)$$

Here, T denotes the temperature in Kelvin, and k stands for the Boltzmann constant. In the context of the distribution of relaxation times (DRT), the relaxation peak corresponds to a unique relaxation process within the conductor, and its associated relaxation time serves as the energy barrier that must be overcome during this conductance process. The activation energy (E_{af}) offers valuable insights into the energy barrier encountered by carriers during their migration. Subsequently, the total ionic conductivity and grain boundary ionic conductivity were calculated using the total resistance and grain boundary resistance, respectively, which were obtained from the intercepts of the grain boundary and grain semicircles with the x -axis (Fig. S11). The activation energy (E_a) is calculated using the Arrhenius method:

$$\sigma = \sigma_0 \exp(-E_a/kT) \quad (2)$$

Here, T denotes the temperature in Kelvin, and k stands for the Boltzmann constant. The activation energy E_a derived from the fitting of conductivity serves as an indicator of the variations in the energy barrier that carriers must jump during their migration. As shown in Fig. S10, the values of E_{af} , total E_a , and grain boundary E_a obtained through fitting the DRT grain boundary relaxation peak, total conductivity, and grain boundary conductivity are presented. The agreement between the values of the activation energies

E_{af} and E_a obtained from fitting the grain boundary relaxation peak and the conductivity provides evidence for the consistency and rationality of the relaxation time (DRT) and Nyquist analysis distributions. Although grain boundaries are the primary factors influencing the total conductivity, the fact that the total conductivity is also influenced by grains leads to slight differences in their activation energies (Since the response data of grain boundaries disappear at high temperatures, the consistency issue is described by the values of the activation energies E_{af} and E_a obtained by fitting the relaxation time and conductivity of grain boundaries). **We added these results and discussions to the revised manuscript.**

Fig. S10 Arrhenius plot corresponding to the bulk relaxation peak (DRT) and Nyquist plot.

4. We would like to express our sincere gratitude to the reviewers for their valuable comments. To enable readers to more clearly understand the specific process of impedance value extraction, we performed fitting using Zview. Meanwhile, we plotted the complete semicircles to allow readers to clearly identify the intercept positions. **We added these results to the revised manuscript.**

Fig. S11 Demonstration of the fitting of the intercepts of the Nyquist semicircles of NCNZ-Na_x ceramics with the x-axis.

5. We sincerely appreciate the reviewers for their insightful comments. The temperature dependence of capacitance was investigated by utilizing the impedance data corresponding to grains and grain boundaries (Fig. S11). For the separate analysis of the grain and grain boundary data, an equivalent circuit composed of two parallel RC elements was employed for fitting analysis. The impedance can be expressed by the following equation:

$$Z' = \left(\frac{1}{R_g} + j\omega C_g \right)^{-1} + \left(\frac{1}{R_{gb}} + j\omega C_{gb} \right)^{-1} \quad (3)$$

$$M^* = j\omega C_0 Z^* \quad (4)$$

The corresponding equations for complex electrical modulus M'' and M' are obtained by combining with eq. (3) and (4) as follows:

$$M' = \frac{C_0}{C_g} \left(\frac{(\omega R_g C_g)^2}{1 + (\omega R_g C_g)^2} \right) + \frac{C_0}{C_{gb}} \left(\frac{(\omega R_{gb} C_{gb})^2}{1 + (\omega R_{gb} C_{gb})^2} \right) \quad (5)$$

$$M'' = \frac{C_0}{C_g} \left(\frac{\omega R_g C_g}{1 + (\omega R_g C_g)^2} \right) + \frac{C_0}{C_{gb}} \left(\frac{\omega R_{gb} C_{gb}}{1 + (\omega R_{gb} C_{gb})^2} \right) \quad (6)$$

Combination of the electric modulus formalisms since each parallel RC element gives rise to a semicircle in the complex plane (M' vs. M'') is shown in Fig. S8a-c. The high frequency arcs of complex plane are associated with $C_g^{-1} + C_{gb}^{-1}$, whereas the low frequency intercepts are C_{gb}^{-1} . The temperature dependences of the grain and grain boundary capacitance values obtained from fitting at various temperatures are shown in Fig. S8d-e. The curve of the grain capacitance value changing with temperature is consistent with the trend of the dielectric spectra in the Fig. 4b-g, while the change of the grain boundary capacitance with temperature remains relatively stable. Meanwhile, the dielectric constant calculated from the grain capacitance is consistent with the values in Fig. 4b-g. These results indicate that the grains are ferroelectric, while the grain boundaries are non-ferroelectric, and they can be distinguished from the capacitance data: the grain boundary effect manifests as temperature-independent capacitance, whereas the grain effect shows significant temperature dependence.

Fig. S8 Temperature dependence of the C of grains and grain boundaries for NCNZ- Na_x ($x=0.96, 1.0, 1.02$) ceramics.

Besides, some modifications were made in the revised manuscript.

- 1) Adequate **References** were cited in the revised manuscript.
- 2) **We polished the whole manuscript carefully again** and tried to avoid any grammatical and formatting errors. All revised contents were marked in red in the revised manuscript.

We greatly appreciate for Editors' enthusiasms and Reviewer's valuable comments, and hope that these corrections can meet them. Thank you very much.

Yours sincerely,

Zhiyong Liu

Tailoring sodium and oxygen mixed-ion conduction in the A-site non-stoichiometric NaNbO_3 -based ceramics

Z. Liu et al.

Although this script contains some interesting results there are many inconsistencies and a severe lack of rigour when it comes to presenting the data and providing information on how it was analysed. I outline my response in more details below and on this basis I have to suggest the script is rejected.

(i) Composition and nomenclature are confusing. No explanation as to why the starting materials include 4% Ca and Ti on the A and B-sites of the lattice, respectively. Also if we start from the given formula $\text{Na}_x\text{Ca}_{0.04}\text{Nb}_{0.96}\text{Zr}_{0.04}\text{O}_3$ then for $x=0.96$ we have a stoichiometric perovskite ABO_3 , yet this is reported as being an Na-deficient sample with Na and O vacancies and this is very confusing. The authors refer (I think) to $x=1$ being stoichiometric yet this would give a formula of $\text{A}_{1.02}\text{B}\text{O}_{3.02}$ and infer either an excess of A-site cations and oxygen ions (or conversely B-site vacancies). Taking the A and O excess model, this begs the question if ABO_3 perovskites are based on close packing of AO_3 layers how can we achieve an excess of A and O sites when no interstitial sites should be available in such a close packed lattice! (Presumably the lattice is therefore B-site deficient). I appreciate that small changes in stoichiometry associated with A/B ratios can have a dramatic effect on the electrical properties of perovskites but this requires rationalisation within the context of the crystal lattice. Late on in the script the authors suggest the presence of the A-site and O-vacancies in $x=0.96$ is associated with volatilisation of Na_2O . This should come much earlier in the script, otherwise, there is substantial confusion about the changes in composition. It might be better for the authors to frame the changes in composition on a solid solution based on a combination of $\text{Na}_{1\pm x}\text{NbO}_3$ and CaZrO_3 (at a level of 4 mol%) perovskites and introduce the fact that there is known non-stoichiometry in NaNbO_3 and that there are additional issues with Na_2O loss during ceramic processing. This would clarify several issues at the outset regarding compositional design and give the reader a clear steer before the results are presented.

(ii) Structural data from laboratory XRD. There is insufficient information given pertaining to how the Rietveld refinements (RR) of the data were performed and of the results obtained. This is important because the positions of the oxygen ions are critical to the arguments about the shape of the NbO_6 octahedra and how that links to ion migration. Furthermore, where do the interstitial Na ions reside and what are the atomic positions? In the methods section it is therefore important that more information on the procedure and strategy used in the RR of the data are provided. I assume ion sites were set to be fully occupied but what were the final (x,y,z) atomic positions of all the ions and what about the isothermal parameters; were they refined or fixed and what are the values? This is important data and must be included in the results. This is especially the case given that XRD is relatively insensitive to light atoms such as O and Na, yet much is made of the final positions of these ions. Furthermore, there are no errors associated with the values given in Table S1 and these need to be included. There is a typographical error in the caption for Fig S3. It should be Na-O not Nb-O.

(iii) Microscopy. I'm not an expert in TEM but this was convincing (to me at least) that $x=0.96$ contained more Na vacancies than $x=1.0$ and the XPS showed there were variations in oxygen content (both Fig 3). Given the heterogeneity in the electrical behaviour (next section) of these

materials and the interpretation provided, it would be good to show some SEM/EDX of the materials to establish the presence of any core-shell distributions within the samples and also EDX profiles across grains to see any variations in composition at the grain boundaries. This intermediate level of microscopic elemental analysis at the micron level is missing and needs to be included to aid the interpretation of the electrical data.

(iv). The electrical characterisation was extensive and performed by Dielectric Spectroscopy (DS), impedance spectroscopy (IS) using DRT and variable P_{O_2} and ion blocking measurements with YSZ; however, in many cases there was a lack of rigour (eg DRT, more later), it was often difficult to see the 'bulk' response in the Z^* plots based on IS (Figs 5,6,7) and the interpretation between DRT and IS seems flawed, the P_{O_2} analysis is confusing and there is no treatment of the electronic contribution in the extraction of the transport numbers for O and Na in equations 4 and 5.

I find the DS data interesting but the authors should indicate if there is any change in T_{max} associated with the P-R transition across the series of samples as this is a major event in these materials. They should also indicate what frequency was used in extracting the temperature for the T to U transition. There are considerable space charge effects at these temperatures and this may influence the values reported.

The IS data are poorly presented, not well analysed and I'm not convinced by the interpretation. It is very difficult to see the bulk arc (called the first arc in the script) at high frequencies in any of the Z^* plots yet this should be central to the analysis. Figures should be expanded to show it in more detail and the extracted R and C values for it presented as a function of temperature instead of a single analysis at 500 C. The extracted capacitance for this arc for 500 C is given as 4 pF/cm which corresponds to a permittivity of $\sim 40-50$. This is an order of magnitude below the 300 -500 observed in the DS data at 500 C in Fig 4. Given that the materials remain ferroelectric at 500 C, then a capacitance value of 4 pF/cm (no errors on this value provided) for the bulk is too low and physically unreasonable for NaNbO₃-based materials. The authors should plot out extracted C values across the full data range and they should show some temperature dependence. By tracking this onto the DRT (Fig 5 c and 6c) with a time constant of 10⁻⁶ s that is associated with this grain response, it appears not to be thermally activated. It remains at this value between 400 to 600 C and this means that the 'assigned' bulk conductivity is not thermally activated!! This cannot be correct. The larger second arc (Fig 5) at 500 C is deconvoluted into two components that are assigned as grain boundary contributions based on C values of 0.17 and 0.14 nF/cm (no errors or showing of the fits based on the equivalent circuit, again illustrating poor rigour in the analysis). These C values correspond to permittivity of ~ 1500 to 2000 which are high but are more consistent with ferroelectric bulk responses as opposed to grain boundary responses. The authors need to plot the temperature dependence of the extracted C values to establish if they are bulk-like (and presumably temperature dependent) or grain boundary-like (presumably with little temperature dependence). These elements are temperature dependent based on the DRT data and as the authors point out show significant changes at 540 C where there is a phase transition. Surely, it makes more sense that these elements associated with the large second arc are bulk (possibly core and shell?) rather than the grain boundaries? A full analysis of extracted R and C values across the full temperature range is required to ensure the analysis is robust and the interpretation is consistent with the physical processes occurring within the materials as they undergo phase changes. At this stage, it's not worth interpreting the P_{O_2} data until the basic IS data are fully analysed.

The DRT is simply provided in a figure and no indication of any fitting parameters are given – again illustrating poor rigour in information pertaining to data analysis.

Figure 9 based on transport numbers versus x makes some sense to me (at least for O). For $x = 0.96$ there is sufficient Na_2O loss to create the level of oxygen vacancies that dominate the conductivity and the presence of this soda loss is clear from TEM/XPS data. For $x = 1.0$ and 1.02 (which actually have excess Na based on the original formula), the loss of soda probably still occurs but samples have much reduced levels of vacancies and therefore are close to being stoichiometric ABO_3 with very low levels of vacancies and therefore ionic transport is low. The authors suggest that the sodium ion transport number is high in Figure 9 for $x = 1.02$ but this might be linked to the lack of treatment of the electronic contribution to the conductivity used in equation 5 to estimate the transport number for Na. I also refer back to my comments linked to composition in perovskites, where are the interstitial sites for A-site cations in the ABO_3 lattice and can they be located via diffraction (no evidence in this script)? I therefore find no-compelling evidence for high levels of Na ion transport in these materials. In addition, the authors often cite NASICON as an example where there is high Na ion migration. This is true but they are based on framework structures where there are vacant sites and it's possible to use these sites to create interstitial ion conduction via selective doping.